# Purification and Characterization of the Protease from *Staphylococcus xylosus* A2 Isolated from Harbin Dry Sausages

**DOI:** 10.3390/foods11081094

**Published:** 2022-04-11

**Authors:** Hui Wang, Jianhang Xu, Baohua Kong, Qian Liu, Xiufang Xia, Fangda Sun

**Affiliations:** College of Food Science, Northeast Agricultural University, Harbin 150030, China; huiwang169@163.com (H.W.); xujianhangneau@163.com (J.X.); kongbh63@hotmail.com (B.K.); liuqian@neau.edu.cn (Q.L.); xxfang524@163.com (X.X.)

**Keywords:** *Staphylococcus xylosus*, protease, purification, characterization, meat protein

## Abstract

The protease generated from *Staphylococcus (S.) xylosus* A2, which was isolated from Harbin dry sausages, was purified and characterized. The molecular weight of the purified protease was approximately 21.5 kDa, and its relative activity reached the highest at pH 6.0 and 50 °C. At pH 4.0–8.0 and temperatures of 20–50 °C, the protease was stable. Its activity was significantly improved by Ca^2+^ and Zn^2+^ ions (*p* < 0.05). The Michaelis constant and maximum velocity of the protease were 2.94 mg/mL and 19.45 U/mL·min, respectively. The thermodynamic parameters analysis suggested that the protease showed better catalytic properties at 40 °C. Moreover, the protease could hydrolyze meat proteins, and obtained hydrolysate is non-cytotoxic to the HEK-293 cells. These findings provide a theoretical basis for understanding the enzymatic characterization of *S. xylosus* A2 protease and its future application in fermented meat products.

## 1. Introduction

Harbin dry sausages are popular in China due to their high protein content, good taste, and unique flavor. Generally, traditional Harbin dry sausages are made from minced raw pork meat mixed with spices and stuffed into a casing, allowed to ferment spontaneously for about 10–15 days in an open environment, relying primarily on endogenous enzymes and “wild” microorganisms derived from the raw material and fermentation environment [1]. Coagulase-negative staphylococci (CNS) were found to be among the most common beneficial strains in dry sausages, and the nitrate reductase produced by CNS during fermentation can not only cause the product to become bright and red, but also reduce fat oxidation in the sausage [2,3]. Moreover, CNS could promote the breakdown of meat protein and fat into peptides, amino acids, and free fatty acids, improving the flavor of dry sausages [4]. Therefore, some CNS have been used as starter cultures in fermented meat products.

During fermentation, CNS, including *Staphylococcus* (*S*.) *xylosus* and *S. carnosus*, may secrete some beneficial metabolites, such as nitrate reductase, protease, catalase, lipase, and superoxide dismutase [5,6], which could improve physical and chemical indexes, antioxidant properties, and the formation of characteristic flavors and aromatic compounds of fermented meat products. The production of nitrate reductase, superoxide dismutase, and catalase not only promotes the production of nitrosomyoglobin to stabilize the color of meat products, but also decomposes hydrogen peroxide to limit lipid oxidation [7]. Cruxen et al. [8] discovered that sausage containing *S. xylosus* LQ3 was lighter, redder, and had less lipid oxidation than the control sample, which they believed is due to the nitrate reductase, catalase, and superoxide dismutase secreted by *S. xylosus* LQ3 during fermentation. Hu et al. [9] discovered that beef jerky inoculated with *S. xylosus* P2 had a more appealing color, odor, texture, taste, and overall acceptability than naturally fermented beef jerky, which mainly depends on the hydrolysis of the chemical components of protease, lipase, and nitrate reductase produced by *S. xylosus* P2 during fermentation. Berardo et al. [10] reported that CNS proteases can hydrolyze myosin into free amino acids and small peptides, promoting the formation of sausage flavor during fermentation. Therefore, protease is considered an important component of fermented meat products.

Proteases are enzymes that can hydrolyze protein into polypeptides or amino acids [11,12]. Aspartic, cysteine, metallo, and serine proteases are the most common types of proteases found in fermented meat products. Microbial proteases are important during the ripening and quality formation of meat products [1,2,3,4]. Chen et al. [13] isolated and identified *S. xylosus* and lactic acid bacteria (LAB) from Harbin dry sausages, and discovered that protease could hydrolyze sarcoplasmic proteins (SP) and myofibrillar proteins (MP) into glutamic acid and alanine as flavor precursors, potentially promoting the formation of the desired flavor. In addition, Sun et al. [14] isolated the LAB protease from Harbin dry sausages and discovered that these extracellular proteases had high activity and can hydrolyze MP and SP. Although there have been many studies on *S. xylosus*, only a few studies on the biochemical characteristics of *S. xylosus* protease exist. In the present study, the extracellular protease from *S. xylosus* A2 isolated from Harbin dry sausages was purified. Kinetic and thermodynamic properties of the protease, as well as its ability to degrade meat protein, were also investigated to understand its biochemical characteristics and future potential application in fermented meat products.

## 2. Materials and Methods

### 2.1. Materials and Reagents

Bovine serum albumin (BSA), casein, trichloroacetic acid (TCA), ethylenediaminetetraacetic acid (EDTA), β-mercaptoethanol, trypsin, papain, Polyacrylamide gel electrophoresis (PAGE) gel, sodium dodecyl sulfate (SDS), Coomassie brilliant blue R-250, 3-(4,5-dimethylthiazol-2-yl)-2,5-6 diphenyltetrazolium bromide (MTT), phosphate-buffered saline (PBS, containing 1.06 mM K2HPO4 and Na2HPO4·7H2O, and 155 mM NaCl; pH 7.4), Dulbecco’s modified eagle medium (DMEM), and Dimethyl sulfoxide (DMSO) were purchased from Solarbio Technology Co., Ltd. (Beijing, China). All of the chemicals were analytically pure.

### 2.2. Microorganism and Protease Production

*S. xylosus* A2 was isolated from Harbin dry sausages and identified as reported by Hu et al. [7]. It was stored on Mannitol Salt Agar plates at 4 °C.

*S. xylosus* A2 was cultured in mannitol salt broth with peptone 3.5 g/L, yeast extract 6.5 g/L, mannitol 5.0 g/L, NaCl 75.0 g/L, K_2_HPO_4_ 5.0 g/L, KH_2_PO_4_ 1.0 g/L, MgSO_4_ 0.4 g/L, and pH 6.0 [12]. A 2% (*v*/*v*) *S. xylosus* A2 culture was fermented at 30 °C with shaking (200 rpm) for 48 h, to produce protease. The fermentation broth was then centrifuged at 5000× *g* at 4 °C for 15 min to collect the supernatant for protease purification [15].

The growth curve of *S. xylosus* A2 over the reaction time of 60 h was measured at 600 nm.

### 2.3. Protease Activity and Concentration Determination

The protease activity was determined by the Folin-phenol method, as described by Sun et al. with slight modifications [1]. The supernatant obtained from the bacterial culture was diluted with 20 mM phosphate buffer at pH 6.0 (buffer 6.0). A 1 mL of diluted enzyme solution was mixed with the 1 mL of 1% casein substrate solution (*w*/*v*), and incubated at 37 °C for 10 min. The reaction was stopped by adding 2 mL of TCA (0.4 M) solution, and was precipitated for 20 min, and was then centrifuged at 5000× *g* for 10 min to collect the supernatant. For the control sample, TCA was added before incubation. Next, 1 mL of the supernatant was mixed with 5 mL of sodium carbonate (0.4 M) and 1 mL of Folin reagent, and was incubated at 40 °C for 20 min. The absorbance of the sample was measured at 680 nm. The amount of protease producing 1 μg of tyrosine per min was described as one unit (U) of activity. The protein concentration was determined using the Biuret method [16].

### 2.4. Proteases Purification

#### 2.4.1. Ammonium Sulfate Fractionation

The crude enzyme solution was purified using 80% ammonium sulfate [14]. The dried ammonium sulfate was milled to a powder, and added to crude protease solution, stirring continually to fully dissolve 80% ammonium sulfate, then incubated at 4 °C for 8 h. The precipitate was centrifuged at 10,000× *g* at 4 °C for 15 min, dissolved using buffer 6.0, and dialyzed overnight. The fraction with good activity was selected for the next step.

#### 2.4.2. Cation Exchange Chromatography

The crude protease obtained after dialysis was applied to a DEAE-Sepharose™ FF column (2.1 cm diameter × 15 cm) [17], which had previously been equilibrated using 50 mM Tris-HCl buffer at pH 6.0 (buffer A). The bound proteins were then eluted using a linear gradient from 0–1.0 M NaCl in 50 mM Tris-HCl buffer at pH 6.0 (buffer B) at the same rate as the unbound proteins were removed with buffer B at a flow rate of 1 mL/min. Finally, the fractions having a 280 nm peak were collected to further analysis.

#### 2.4.3. Gel-Filtration Chromatography

The protease solution was added into a Sephadex G-75 gel filtration column (1.8 cm diameter × 40 cm), eluted with buffer A, and 4 mL fractions were collected into a tube. The fractions having a 280 nm peak were collected to freeze-dried and stored at 4 °C.

### 2.5. SDS-PAGE

SDS-PAGE was performed with a 12% resolving gel and a 5% stacking gel to determine the purity and molecular weights of the purified protease [18]. The prepared gel was stained with 0.25% Coomassie blue R-250 for 15 min.

### 2.6. Effects of pH and Temperature on the Protease Activity and Stability

To estimate the optimum pH, the protease was incubated with 1% casein solutions in different buffers, including sodium citrate buffer (0.2 M): pH 3.0–5.0; potassium phosphate buffer (0.2 M): pH 6.0–9.0; and glycine-NaOH buffer (0.2 M): pH 9–11. The protease was incubated in different buffers at 37 °C for 0–100 min with an interval of 20 min to examine the protease activity.

The protease solution was incubated with 1% casein solutions at pH 6 and temperature from 20 °C to 80 °C to evaluate the optimum temperature. The protease solution was incubated at different temperatures (20–80 °C) for 0–100 min with an interval of 20 min and measured the protease activity. The maximal activity of protease was considered to be 100%.

### 2.7. Effects of Metallic Ions and Inhibitors on Protease Activity

At final concentrations of 1 mM and 10 mM, different metallic ions and inhibitors were added to protease solutions and incubated at 50 °C and pH 6.0 for 30 min. The metallic ions and inhibitors used were: Fe^3+^, Fe^2+^, Cu^2+^, Zn^2+^, Mg^2+^, Ca^2+^, Na^+^, K^+^, EDTA, and β-mercaptoethanol. The control sample was protease without metallic ions or inhibitors (100%). The residual enzyme activity was tested.

### 2.8. Determination of Activation Energy

The activation energy (*E_a_*) was determined according to the slope of the Arrhenius plot [19] at 20 °C to 50 °C, with the following formulas:(1)$$E_{a}=-\;\mathrm{slope}\;\times \;R$$
where: *R* and *T* are the gas constant and absolute temperature, respectively.

### 2.9. Determination of Kinetic and Thermodynamics Parameters of the Protease

The kinetic constants of purified *S. xylosus* A2 protease was determined by observing the reaction rate between the protease and casein as substrates (1, 2, 2.5, 4, 5, 10, 15, and 20 mg/mL) at 40 °C and pH 6.0 for 10 min. The Michaelis constant (*K_m_*) and maximum velocity of the reaction (*V_max_*) were estimated using Line weaver Burk plots [15]. The enthalpy (Δ*H**), entropy (Δ*S**), and free energy (Δ*G**) activation were determined using the following formulas [20]:(2)$$\Delta H^{*}{=E}_{a}-\;RT$$
(3)$$\Delta S^{*}=\;(\Delta H^{*}-\Delta G^{*})/T\;$$
(4)$$\Delta G^{*}=-RTln(\frac{k_{cat}h}{k_{b}}\;\times \;T)\;$$
(5)$$k_{cat}=(\frac{k_{b}T}{h}{)\;\times \;e}^{(-\frac{\Delta H^{*}}{RT})}{\times \;e}^{\left(\frac{\Delta S^{*}}{R}\right)}$$
where *k_b_* is the Boltzmann’s constant (R/N), and *h* is the Planck’s constant.

### 2.10. Thermal Inactivation

The activity of the protease solution was measured at 40 °C, 50 °C, and 60 °C, as indicated in Section 2.3. The half-life period (*T*_1/2_) is the time required for a 50% reduction in the initial activity, and the D value is the time required for a 90% reduction in the initial activity at a certain temperature. The inactivation constant (*K_d_*), *T*_1/2_, and D value were calculated by the following formulas [21]:(6)$$T_{1/2}{=ln2/K}_{d}$$
(7)$$D=2{.303/K}_{d}$$

Moreover, *E*_*a*(*d*)_ was calculated according to the correlation coefficient (slope) of ln*K_d_* against 1000/T. The thermal inactivation parameters (Δ*H*_d_*, Δ*S*_d_*, and Δ*G*_d_*) of the protease were calculated by Equations (8)–(11), respectively.
(8)$$K_{d}=(\frac{k_{b}T}{h}{)\times e}^{(-\frac{\Delta H_{d}^{*}}{RT})}{\times e}^{\left(\frac{\Delta S_{d}^{*}}{R}\right)}$$
(9)$$\Delta H_{d}^{*}{=E}_{a(d)}-\;RT$$
(10)$$\Delta S_{d}^{*}=(\Delta H_{d\;}^{*}-\Delta G_{d}^{*})/T$$
(11)$$\Delta G_{d}^{*}=-RTln(\frac{K_{d}h}{k_{b}}\times T)\;$$

### 2.11. Extraction and Degradation of MP and SP

The MPs and SPs were extracted from lean pork by the report of Sun et al. [5]. The MPs and SPs obtained were kept in an ice bath until use. MP and SP concentrations were diluted to 10 mg/mL using buffer 6.0. MP and SP solutions were then mixed with the same volume of *S. xylosus* A2 protease, trypsin, and papain solution (40 U/mL, pH 6.0), respectively, and incubated in a shaking (100 rpm) water bath at 37 °C for up to 120 min (30, 60, 90 and 120 min) to investigate the evolution of hydrolysis with time by SDS-PAGE. In addition, 5 mL of MP and SP solution was mixed with 5 mL of buffer 6.0, respectively, which was native MP and native SP (0 min of hydrolysis).

### 2.12. Determination of Particle Size

A Mastersizer (Malvern Instruments Ltd., Worcestershire, UK) was used to measure the mean particle size (*d*_(4,3)_) and polydispersity index (PDI) of the protein solution.

### 2.13. Confocal Laser Scanning Microscopy (CLSM)

The distribution and size of meat proteins were examined by a CLSM (Leica TCS SP5, Heidelberg, Germany). To stain the protein phase, a 20 μL of 0.1% Nile blue solution (in water) was mixed evenly with 1.0 mL of protease-treated protein solution. To observe the proteins, the stained protein solution (10 μL) was coated on clean concave microscope slides with cover slips using a 40 × HCPL APO/20 × oil immersion objective [22] and a 633 nm helium-neon laser.

### 2.14. Cell Toxicity Assay

The MP and SP solution treated with *S. xylosus* A2 protease (0, 60, and 120 min) was heated at 100 °C for 5 min to inactivate the protease, and was centrifugated at 2000× *g* for 10 min at 4 °C to obtain the supernatant.

The cytotoxicity of obtained supernatant was studied in the human nontumorigenic embryonic kidney cell line (HEK-293 cells) purchased from Chinese Academy of Sciences using MTT reduction assay. A 20 μL of obtained supernatant was added into the HEK-293 cells solution in sextuplicate and incubated for 24 h. After incubation, the HEK-293 cells were washed with PBS twice and 200 μL of MTT solution was added, and incubated for 4 h. The MTT solution was removed, 200 μL of DMSO was added and gently shaken for 15 min to dissolve the formazan crystals. Untreated HEK-293 cells were used as the control and its cell viability was expressed as 100%. The absorbance of treated cells solution was measured at 490 nm using a microplate reader (Bio-Rad Inc., Hercules, CA, USA). The percentage of cell viability was calculated using the following formulas: (12)$$\mathrm{Cell}\;\mathrm{viability}\;(\%)=\frac{{\mathrm{OD}}_{2}-{\;\mathrm{OD}}_{0}}{{\mathrm{OD}}_{1}-{\;\mathrm{OD}}_{0}}\times 100\%$$
where OD_0_, OD_1_ and OD_2_ refer to the absorbance of the culture medium, the control, and treated cells at 490 nm.

### 2.15. Cell Morphology

An 1800 μL of the HEK-293 cells were seeded into 6-well plates (300,000 cells/well) and incubated for 12 h. The obtained supernatant (200 μL) in Section 2.14 was added to the HEK-293 cells and incubated for 24 h. Afterward, the HEK-293 cells were washed with PBS twice. The morphological change of treated cells was visualized using an inverted microscope.

### 2.16. The Peptide Concentration Measurements

The peptide concentration in protein solution was measured with minor modifications. A 5 mL TCA (15% *w*/*v*) solution was added to each of the 5 mL protease-treated MP and SP solutions, mixed evenly, and allowed to react for 30 min and centrifuged at 5000× *g* for 10 min [23]. The absorbance of the supernatant at 540 nm was tested using the Biuret method.

### 2.17. Statistical Analysis

Three independent experiments were carried out. All measurements were conducted in triplicate. IBM SPSS software version 21 (SPSS Inc, Chicago, MI, USA) was used to analyze the data, which was recorded as means ± standard errors (SE, *n* = 3 for all treatments). The significance of the main effects (*p* < 0.05) was analyzed using one-way analysis of variance and Duncan’s multiple range tests.

## 3. Results and Discussion

### 3.1. The Growth Curve of S. xylosus A2

The growth curve of *S. xylosus* A2 with the extension of fermentation time is shown in Figure 1. With the extension of fermentation time, the OD value of *S. xylosus* A2 increased gradually. At 18 h of fermentation, the OD value of *S. xylosus* rapidly reached a 1.52, indicating that this stage was the logarithmic growth stage of the *S. xylosus*; at 24 h of fermentation, its OD value increased to 1.78, while it almost plateaued at 1.81 after 30 h of fermentation, which showed that the *S. xylosus* entered a stable growth period.

### 3.2. Protease Purification

The result of 80% ammonium sulfate saturation is shown in Table 1. Purification fold and yield were approximately 1.15 and 51.4%, respectively, and obtained the crude protease solution, which had a specific activity of 2.69 U/mg.

The DEAE-Sepharose™ FF column result is shown in Figure 2A. During the elution from the DEAE-Sepharose™ FF column, four protein absorption peaks emerged, among which one peak (fractions 28–33) demonstrated the highest protease activity. Therefore, the protein solution from fractions 28–33 was pooled and loaded onto a gel filtration column (Figure 2B). Fractions 15–27 exhibited the highest protease activity and were pooled for analysis of *S. xylosus* protease biochemical characteristics. After the two-step purification, the specific activity, purification fold, and yield were 33.3 U/mg, 14.23, and 14.7%, respectively (Table 1). In addition, only a band with molecular weights of 21.5 kDa was observed after gel-filtration purification in the SDS-PAGE (Figure 2B), indicating that the purified protease from *S. xylosus* A2 was a single protein.

### 3.3. Effects of pH and Temperature on the Protease Activity and Stability

The effect of pH on *S. xylosus* A2 protease activity is shown in Figure 3A. The protease activity increased from pH 3 to 6 and decreased from pH 6 to 11, with the maximum activity at pH 6.0, which was consistent with the *Lactobacillus fermentum* R6 protease and *Lactobacillus curvatus* R5 protease [14,24]. The pH stability profile (Figure 3B) showed that the protease relative activity gradually decreased at pH 3–11 as the incubation time increased. After incubation at pH 4–8 for 100 min, more than 70% of the original activity was retained, demonstrating that the *S. xylosus* protease exhibited good stability at pH 4–8, which is an important feature of *S. xylosus* protease in the application of fermented meat products [5].

The optimal protease activity was at 50 °C (Figure 3C). Its activity was more than 60% of the initial activity at 20–60 °C, but decreased drastically as the temperature increased, similar to that of *Micrococcus* sp. alkaline protease [25]. In addition, the activity of the *S. xylosus* A2 protease at 20–70 °C gradually decreased with increasing incubation time (Figure 3D). After 100 min, the relative activity of the *S. xylosus* A2 protease was above 70% at 20–40 °C and still above 50% at 50 °C. After 40 min, the relative activity of the *S. xylosus* A2 protease decreased rapidly to 30% of the original activity at 60 °C, and the protease activity of the *S. xylosus* A2 protease was almost 0 at 70 °C, which was due to the high temperature breaking some non-covalent bonds of the protease [17,26], resulting in conformational changes. These results showed that the protease from *S. xylosus* A2 exhibited high thermal stability at 20–50 °C.

### 3.4. Influence of Metallic Ions and Inhibitors on Activity of the S. xylosus A2 Protease

The influence of metallic ions and inhibitors on the activity of the *S. xylosus* A2 protease is shown in Figure 4A. At 10 Mm, Na^+^, K^+^, and Mg^2+^ had significant effect on *S. xylosus* A2 protease activity (*p* < 0.05). Ca^2+^ and Zn^2+^ at 1 mM and 10 mM significantly improved (*p* < 0.05) the activity of the protease. Ca^2+^ and Zn^2+^ at high concentration increased enzyme activity by 125.09% and 120.03%, respectively, which was probably a result of Ca^2+^ and Zn^2+^ promoting the binding of the protease active sites to casein [27]. Benkiar et al. [28] observed that Ca^2+^ and Zn^2+^ could enhance the relative protease activities by 450% and 180%, respectively. However, the protease activity was inhibited by Cu^2+^, Fe^2+^, and Fe^3+^, as observed in *Lactobacillus curvatus* R5 protease and *Bacillus* sp. ZJ1502 protease [12,14]. *S. xylosus* A2 protease activity was decreased to 55.37% and 40.07% by 1 mM and 10 mM Cu^2+^, respectively, possibly since Cu^2+^ caused nonspecific binding and aggregation of proteases [25]. At high concentration, the protease activity was slightly inhibited by EDTA, whereas at low concentrations, its activity was significantly inhibited by EDTA. Furthermore, β-mercaptoethanol significantly (*p* < 0.05) inhibited protease activity, which could be due to β-mercaptoethanol reducing the disulfide bond of protease [26]. In this study, the addition of Ca^2+^ significantly enhanced *S. xylosus* A2 protease activity, which is consistent with the findings of Reddy et al. [27], indicating that the protease contains a Ca^2+^ binding site(s).

### 3.5. Ea Analysis

*E_a_* is the energy required to generate an enzyme-substrate complex [29]. The activities of *S. xylosus* A2 protease were measured at 20–50 °C. According to Arrhenius plots (Figure 4B), the *E_a_* was approximately 13.80 kJ·mol^−1^ (Table 2), which was lower 62 kJ·mol^−1^ of serine protease from *Aspergillus fumigatus* [30], 17.31 kJ·mol^−1^ of alkaline protease from *Bacillus stearothermophilus* [31], and 92.7 kJ/mol of protease from *Bacillus* sp. P45 [32]. In general, the low *E_a_* value indicated the good binding of the enzyme to the substrate [31].

### 3.6. Kinetic Constants and Thermal Stability of Casein Hydrolysis

Lineweaver–Burk plots were used to study the kinetic parameters of casein hydrolysis at pH 6.0 and 50 °C (Figure 4C). The results in Table 2 showed that the *K_m_*, *V_max_*, and *k_cat_* (catalytic constant) values of the protease from *S. xylosus* were 2.94 mg·mL^−1^, 19.45 mg·min^−1^, and 3.24 s^−1^, respectively, for the substrate casein. *K_m_* refers to the affinity of an enzyme to the substrate. A lower *K_m_* value suggests a higher affinity of an enzyme to the substrate [17]. The *Micrococcus* sp. protease had a high *K_m_* value of 6.39 mg·mL^−1^ for the substrate casein [25]. The protease from Withania coagulans fruit had a low *K_m_* of 1.29 mg·mL^−1^ for the substrate casein [33]. These results showed that different protease sources exhibited different affinities for casein.

Table 2 shows the thermodynamic parameters of the *S. xylosus* protease. For the *S. xylosus* protease, Δ*H** values of 11.11 kJ·mol^−1^, Δ*G** values of 45.16 kJ·mol^−1^ and Δ*S** values of 105.42 J·mol^−1^K^−1^ were calculated at 30 °C. Additionally, Δ*H** values of 31.62 kJ·mol^−1^, Δ*G** values of 94.99 kJ·mol^−1^, and Δ*S** values of −184.75 J·mol^−1^K^−1^ were reported for an alkaline protease from *Nocardiopsis dassonvillei* strain OK-18 [34]. The lower Δ*H** and Δ*S** values of *S. xylosus* A2 protease indicated that the transition state or enzyme-to-substrate complex was formed easily. Additionally, the lower Δ*G** value suggested in a transition state, it was easier to convert an enzyme-substrate complex into a product [21].

Based on an Arrhenius plot (Figure 4D), the activation energy of the *S. xylosus* protease denaturation of 28.52 kJ·mol^−1^ (R^2^ = 0.98) was calculated (Table 3). This value was lower 32.79 kJ·mol^−1^ of alkaline protease from *Bacillus licheniformis* [19] and 28.8 kJ·mol^−1^ of protease from *Aspergillus tamarii* [21]. A lower *E_(a)d_* value indicated lower thermal stability in the *S. xylosus* protease.

Moreover, the thermodynamic parameters for irreversible thermal inactivation were calculated at 40 °C, 50 °C, and 60 °C, as shown in Table 3. The Δ*H*_d_* was defined as the total energy required to denature the enzyme. The higher the value, the higher the energy required for the thermal denaturation of the enzyme [35]. The Δ*H*_d_* values of the *S. xylosus* protease were 25.92, 25.83, and 25.75 kJ/mol at 40 °C, 50 °C, and 60 °C, respectively, indicating a high energy requirement for thermal denaturation of the protease at 40 °C. The Δ*G*_d_* represents the spontaneity of thermal unfolding inactivation processes and is a critical thermodynamic parameter for determining enzyme stability [21]. The *S. xylosus* protease has the highest Δ*G*_d_* value at 60 °C, which was 93.88 kJ/mol. High and positive Δ*G*_d_* values prove that the heat denaturation reaction of protease is non-spontaneous [21].

Table 3 shows the thermal stability results of the *S. xylosus* protease at 40 °C, 50 °C, and 60 °C. As temperature increases, *T*_1/2_ gradually decreases, whereas the inactivation constant (*K_d_*) increases progressively. Furthermore, as temperature increased, the D value of the *S. xylosus* protease followed the same trend as the *T*_1/2_. The thermal stability of *S. xylosus* protease was the highest at 40 °C (*T*_1/2_ = 102.64 min; *K_d_* = 0.0067; D = 343.73 min), followed by 50 °C (*T*_1/2_ = 77.02; *K_d_* = 0.009; D = 255.89), and the lowest at 60 °C (*T*_1/2_ = 52.89 min; *K_d_* = 0.013; D = 177.15), indicating that irreversible denaturation of the *S. xylosus* protease became increasingly noticeable. This could be attributed to the destruction of some active protease sites by high temperatures, which accelerate the irreversible denaturation of the *S. xylosus* protease [35].

### 3.7. Protein Profiles

Proteolysis is a critical process in determining the quality and flavor of Harbin dry sausages [5]. It was important to investigate the degradation effects of *S. xylosus* A2 protease by testing its ability to hydrolyze meat protein. In recent years, trypsin and papain were the most widely used commercial proteases in meat industry. Therefore, in this study, SDS-PAGE was used to observe the hydrolysis characteristics of purified *S. xylosus* A2 protease, trypsin, and papain on meat proteins. The hydrolysis characteristics of MP and SP by *S. xylosus* A2 protease were understood by comparing the hydrolysis results of trypsin and papain. As shown in Figure 5A, the degree of MP hydrolysis by *S. xylosus* A2 protease gradually increased with the extension of time, suggesting that the protease from *S. xylosus* A2 could promote MP hydrolysis. MPs from pork have six major proteins, which include myosin heavy chain (MHC, 220 kDa), paramyosin (97.2 kDa), actin (45 kDa), tropomyosin (TM, 40 kDa), troponin (36 kDa), and myosin light chain (MLC, 17–20 kDa). Throughout the hydrolysis, the paramyosin and tropomyosin were significantly degraded, and the intensities of the MHC, actin, and troponin bands gradually decreased, accompanied by the emergence of new proteins (<66.4 kDa). Trypsin (Figure 5C) and papain (Figure 5E) hydrolyze MP in a similar way to *S. xylosus* A2 protease. Moreover, *S. xylosus* A2 protease may promote SP hydrolysis (Figure 5B). The SP has four major protein bands on the SDS-PAGE electrophoretogram, including phosphorylase, creatine kinase-M type, glyceraldehyde dehydrogenase, and myoglobin. The phosphorylase band disappeared almost completely after 30 min of hydrolysis, but the creatine kinase-M type and glyceraldehydes dehydrogenase bands gradually disappeared over time. Similar results are obtained when SP is degraded by papain (Figure 5F). Trypsin (Figure 5D) could only degrade phosphorylase and glyceraldehyde dehydrogenase, unlike protease and papain. These results demonstrated that *S. xylosus* A2 protease could accelerate the degradation of meat protein and also promoted the production of new protein components, indicating that the *S. xylosus* protease has potential application value in fermented meat products.

### 3.8. Distribution and Particle Diameter

The *d*_(4,3)_, PDI, and CLSM images of meat proteins (MP and SP) treated with *S. xylosus* A2 protease are shown in Figure 5. The native MP (Figure 6A) and SP (Figure 6B) have the intact fibrous structures and relatively large average particle size, as reported by Liu et al. [22], who analyzed the structure and morphology of pork MP in water. The intact fibrous structure of MPs and SPs treated with *S. xylosus* A2 protease was gradually degraded into small particles and dispersed uniformly in the solution with the prolongation of hydrolysis time, which was likely due to *S. xylosus* A2 protease breaking the chemical bonds between protein-protein, reducing MP and SP particle diameter and inhibiting meat protein aggregation [36]. After hydrolysis (120 min), the *d*_(4,3)_ and PDI in MP treated with *S. xylosus* A2 protease reduced from 61.56 µm to 21.49 µm and from 3.39 µm to 2.32 µm, respectively, and the *d*_(4,3)_ and PDI in SP treated with *S. xylosus* A2 protease reduced from 45.18 µm to 17.64 µm and from 4.05 µm to 1.42, which was consistent with the CLSM images results. This result showed that *S. xylosus* A2 protease could effectively disperse meat protein aggregates.

### 3.9. Cell Toxicity Assay

The toxicity of hydrolyzed products of meat proteins treated with protease are essential factors that must be considered to ensure their safety. The HEK-293 cells are the immortalized human embryonic kidney cell and the main remodeling cells of the kidney. The adrenal glands are quite important endocrine organs of the human body, which plays an extremely important role in maintaining normal life activities of the human body, and they also have characteristics of rapid reproduction and easy culture [37,38]. Therefore, in this study, the HEK-293 cells were used to evaluate the toxicity of meat protein (MP and SP) treated with *S. xylosus* A2 protease, as shown in Figure 7A. As the hydrolysis time increases, the viability of treated HEK-293 cells increased significantly (*p* < 0.05), which may be due to the fact that the hydrolysates of MP and SP treated with the *S. xylosus* A2 protease improve the environment in which HEK-293 cells grow. At 120 min, the cell viability was 112% (MP treated with *S. xylosus* A2 protease) and 119% (SP treated with *S. xylosus* A2 protease), respectively, indicating that MP and SP treated with *S. xylosus* A2 protease exhibited no cytotoxicity against the HEK-293 cells.

### 3.10. Morphological Features of Cell

To visualize the morphological features of HEK-293 cells in absence or presence of meat protein treated with *S. xylosus* A2 protease, we use inverted microscope technique, as shown in Figure 7B. Untreated and treated HEK-293 cells grew well and displayed excellent health characteristics, which indicated that MP and SP treated with *S. xylosus* A2 protease could not induce typical morphological features of HEK-293 cell.

### 3.11. Peptide Concentrations

The solubility of peptides was shown to be highly correlated with the NaCl concentration and pH of the solution [23]. Hence, a solution with a pH of 6.0 and no NaCl was chosen as the hydrolysis condition for the determining peptide concentration. As shown in Figure 8, protease-treated MP (Figure 8A) and SP (Figure 8B) solutions yielded higher peptide concentrations than native MP (0 min) and SP (0 min), whereas their peptide concentrations increased gradually (*p* < 0.05) with the prolongation of hydrolysis time, which could be since *S. xylosus* A2 protease decreased the *E_a_* required for the conversion of MPs and SPs to peptides, accelerating the breakdown of polypeptide chains and thus increasing peptide concentration in the solutions [39]. This result suggested that the *S. xylosus* A2 protease could accelerate the hydrolysis of MP and SP into small peptides, which is an important factor in the formation of aroma compounds and their precursors. Therefore, the *S. xylosus* A2 protease has potential application value in fermented meat products.

## 4. Conclusions

In this study, an extracellular protease from *S. xylosus* A2 was purified, and its biochemical characteristics were investigated. The protease exhibited high activity at pH 6.0 and 50 °C. Na^+^, K^+^, Mg^2+^, Ca^2+^, and Zn^2+^ significantly (*p* < 0.05) improved its activity. The *S. xylosus* protease exhibited excellent thermostability at low temperature (<40 °C), while its activity decreased rapidly at high temperature (>50 °C), indicating that the protease was not suitable for working at high temperature. Furthermore, the protease exhibited a good ability to degrade meat proteins, and obtained hydrolysate is safety. In summary, this study provided a theoretical basis for understanding the enzymatic properties of *S. xylosus* A2 protease and its future application in traditional fermented meat products (<30 °C). In our further research, the *S. xylosus* A2 protease will be added to traditional fermented meat products to investigate the quality and flavor of products.

## Figures and Tables

**Figure 1 foods-11-01094-f001:**
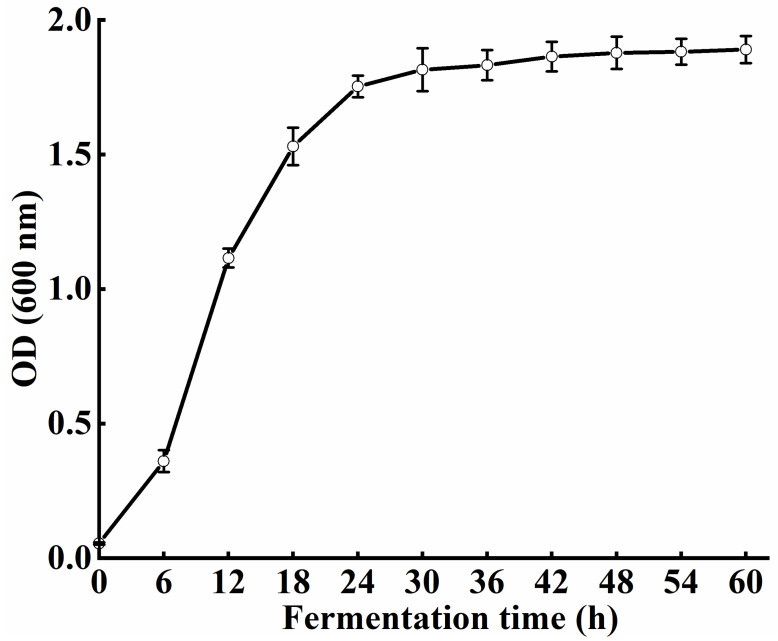
The growth curve of the *S. xylosus* A2.

**Figure 2 foods-11-01094-f002:**
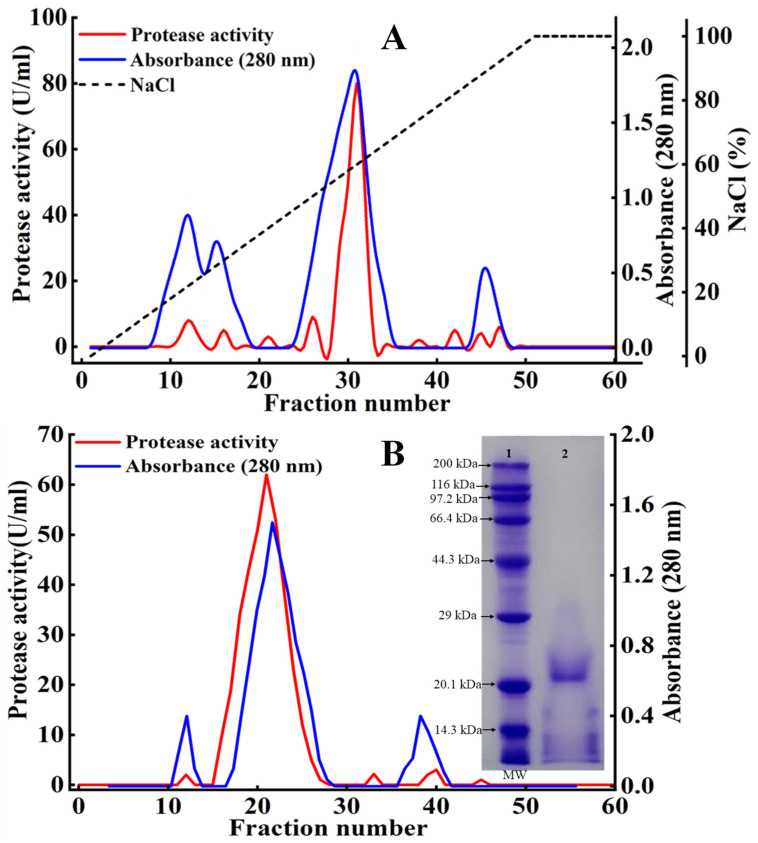
The *S. xylosus* A2 protease purification by cation exchange chromatography (**A**) and gel-filtration chromatography (**B**) and SDS-PAGE analysis of the sample from gel filtration. Lane 1, protein molecular weight marker; lane 2, the sample after gel-filtration chromatography purification.

**Figure 3 foods-11-01094-f003:**
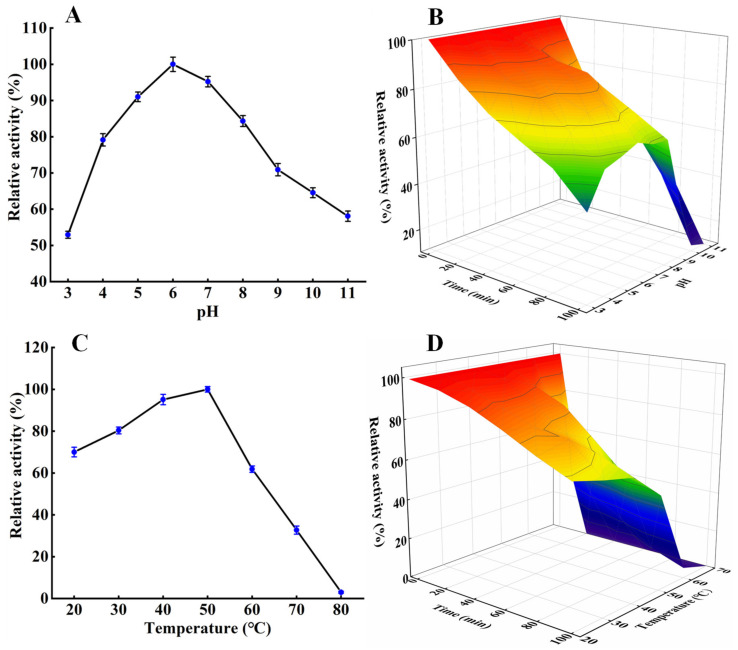
Effect of pH on protease activity (**A**) and stability (**B**); effect of temperature on protease activity (**C**) and stability (**D**).

**Figure 4 foods-11-01094-f004:**
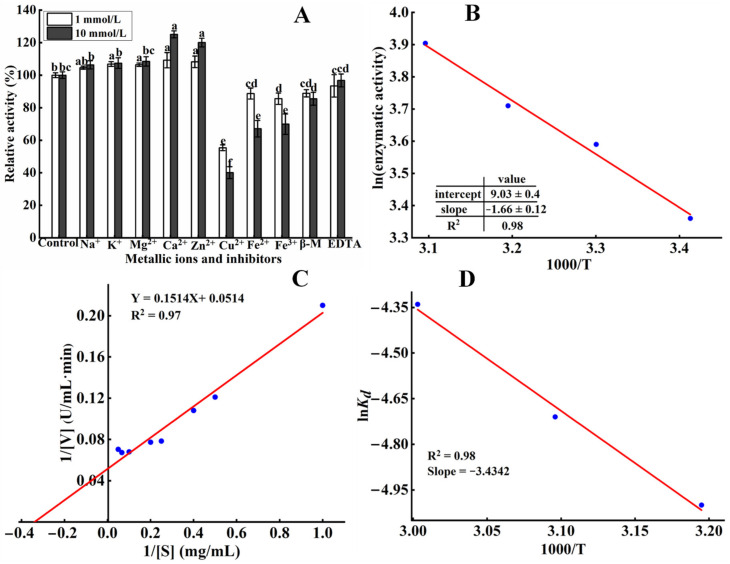
(**A**) Influence of metallic ions and inhibitors on the *S. xylosus* A2 protease activity. Different letters in the same column represent significant differences in different samples (*p* < 0.05). (**B**) Arrhenius plot of the *S. xylosus* A2 protease. (**C**) Lineweaver-Burk plot for the *S. xylosus* A2 protease using casein as the substrate. (**D**) Arrhenius plots for the thermal denaturation (*E_a(d)_*) of the *S. xylosus* A2 protease.

**Figure 5 foods-11-01094-f005:**
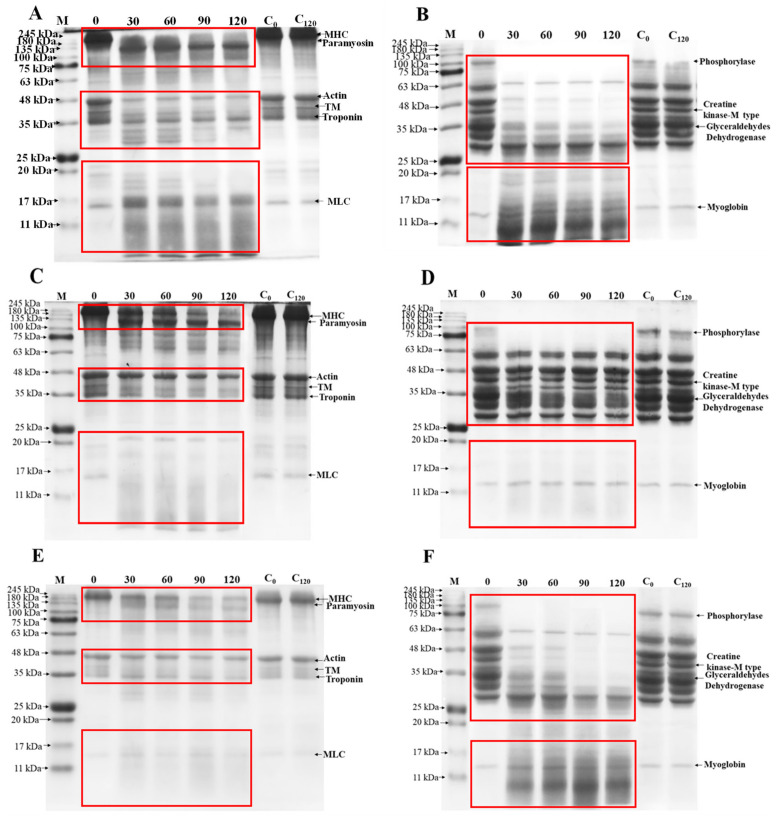
Enzymatic hydrolysis of myofibrillar (**A**) and sarcoplasmic proteins (**B**) by the *S. xylosus* A2 protease. Enzymatic hydrolysis of myofibrillar (**C**) and sarcoplasmic proteins (**D**) by trypsin. Enzymatic hydrolysis of myofibrillar (**E**) and sarcoplasmic proteins (**F**) by papain. C_0_ and C_120_ refer to myofibrillar and sarcoplasmic proteins incubated with ultrapure water for 0 and 120 min under the same conditions. M refers to the molecular weight of the protein standard; MHC refers to myosin heavy chain; TM refers to tropomyosin; MLC refers to myosin light chain.

**Figure 6 foods-11-01094-f006:**
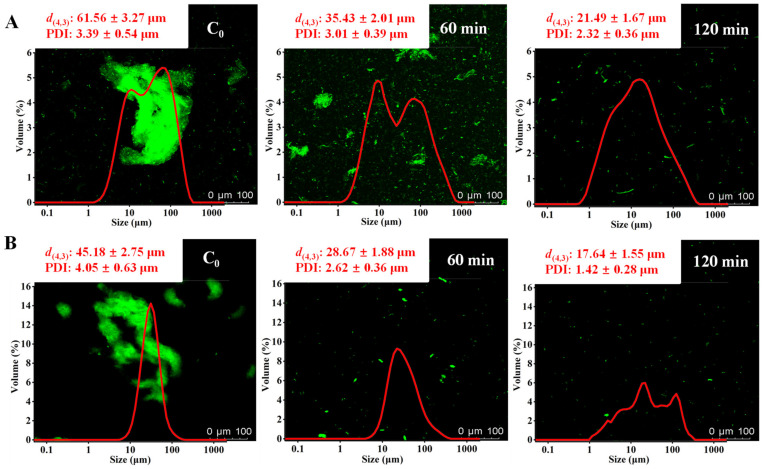
Confocal laser scanning microscopy (CLSM) micrographs and particle distribution of myofibrillar (**A**) and sarcoplasmic proteins (**B**) treated with the *S. xylosus* A2 protease. *d*_(4,3)_ and PDI represent the volume average particle size and polydispersity index, respectively.

**Figure 7 foods-11-01094-f007:**
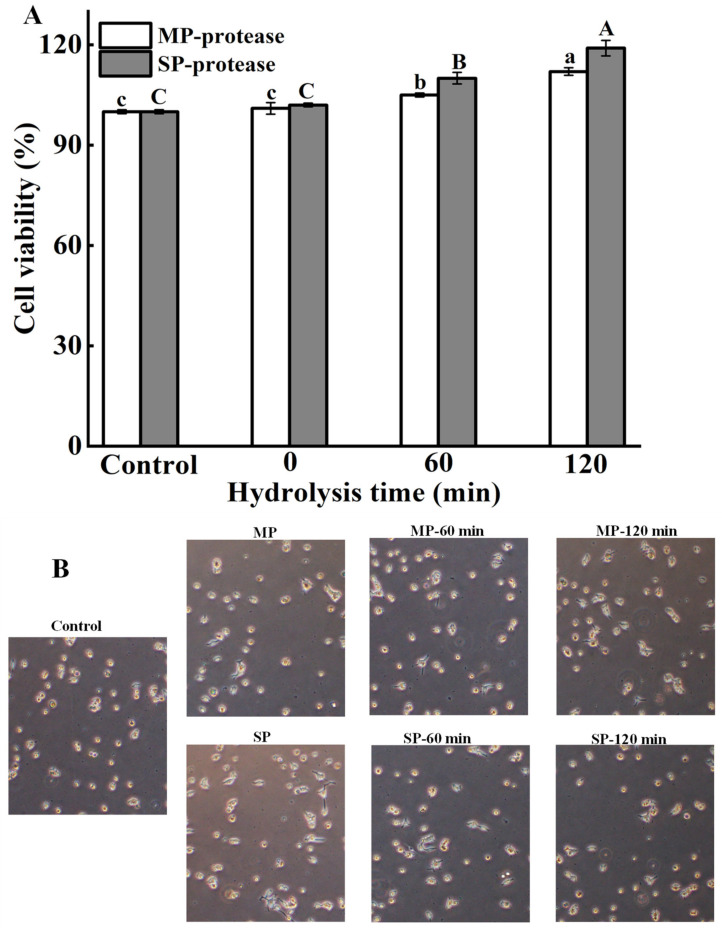
(**A**) Cytotoxicity assay of HEK-293 cells treated with MP and SP. (**B**) Morphology of HEK-293 cells treated with MP and SP. Different letters in the same column represent significant differences in different samples (*p* < 0.05).

**Figure 8 foods-11-01094-f008:**
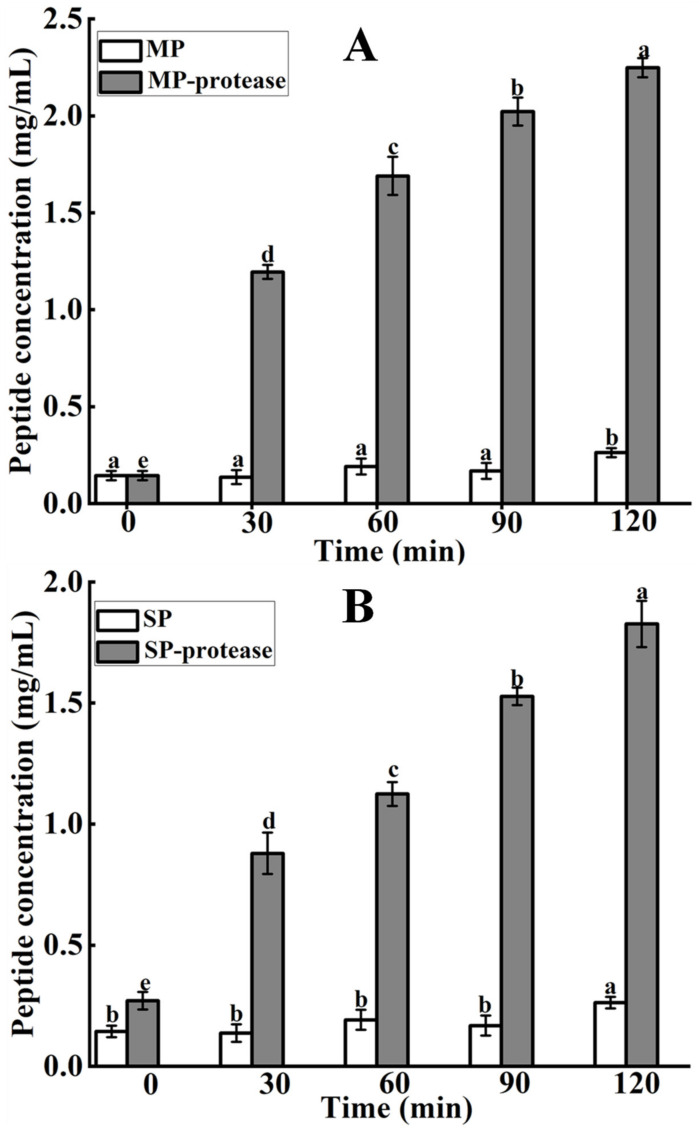
The peptide concentrations of myofibrillar (**A**) and sarcoplasmic proteins (**B**) treated with the *S. xylosus* A2 protease. Different letters in the same column represent significant differences in different samples (*p* < 0.05).

**Table 1 foods-11-01094-t001:** Purification results of the protease from *S. xylosus* A2.

Purification Steps	Total Activity (U)	Total Protein (mg)	Specific Activity (U/mg Protein)	Purification Fold	Yield (%)
Culture broth	68.23 ± 1.26	29.23 ± 1.10	2.34 ± 0.21	1	100
Ammonium sulfate precipitation	35.54 ± 1.06	13.24 ± 0.36	2.69 ± 0.30	1.15	51.4
DEAE-Sepharose™ FF	19.34 ± 1.11	1.25 ± 0.04	15.81 ± 0.56	6.75	27.9
Sephadex-G75 chromatography	10.36 ± 1.20	0.33 ± 0.02	33.31 ± 0.68	14.23	14.7

Specific activity = Total activity (U)/Total protein (mg); Yield (%) = Total activity/Total activity of culture medium supernatant × 100; Purification fold = Specific activity/Specific activity of crude extract.

**Table 2 foods-11-01094-t002:** Kinetic and thermodynamic parameters of the *S. xylosus* A2 protease.

Parameter	Data
*E_a_* (kJ·mol^−1^)	13.80 ± 0.81
*V_max_* (mg·min^−^^1^)	19.45 ± 0.69
*K_m_* (mg·mL^−^^1^)	2.94 ± 0.14
*k_cat_* (s^−1^)	3.24 ± 0.29
Δ*H** (kJ·mol^−1^)	11.11 ± 0.45
Δ*G** (kJ·mol^−1^)	45.16 ± 1.48
Δ*S** (J·mol^−1^K^−1^)	−105.42 ± 2.17

*k_cat_* = *V_max_*/[*e*] where [*e*] = 0.1 μmol/mL (enzyme concentration).

**Table 3 foods-11-01094-t003:** Thermodynamic parameters for irreversible inactivation of the *S. xylosus* A2 protease.

Temperature	T (°K)	*T*_1/2_ (min)	*K_d_* (min^−1^)	*D* (min)	Δ*H*_d_* (kJ·mol^−1^)	Δ*G*_d_* (kJ·mol^−1^)	Δ*S*_d_* (J·mol^−1^K^−1^)	*E*_(*a*)*d*_ (kJ·mol^−1^)
40 °C	313	102.64 ± 2.88	0.0067 ± 0.0004	343.73 ± 25.63	25.92 ± 1.38	89.80 ± 4.11	−204.09 ± 10.67	-
50 °C	323	77.02 ± 1.71	0.009 ± 0.0003	255.89 ± 17.37	25.83 ± 2.61	91.97 ± 5.63	−204.77 ± 8.83	28.52 ± 2.13
60 °C	333	52.89 ± 1.95	0.013 ± 0.0072	177.15 ± 16.74	25.75 ± 1.93	93.88 ± 4.31	−204.59 ± 6.57	-

## Data Availability

The data presented in this study are available within the article.

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
