# Peer review of "Purification and Characterization of the Protease from *Staphylococcus xylosus* A2 Isolated from Harbin Dry Sausages"

_foods, 2022, doi:10.3390/foods11081094_

Round 1

Reviewer 1 Report

The current study highlighted purification and characterization of a protease of 21.5 kDa from Staphylococcus (S.) xylosus A2. The enzyme was found stable at 50 ℃ with improved activity in the presence of certain ions. The paper is simply written and easy to understand. However, the major concern of this enzyme is the optimum activity found at 40℃, while many food processing industries used a temperature above 50℃.  How the author justify the application of this enzyme in industry. The manuscript needs a major revision at this step. I have few comments below;

 Comment 1 Line86: “protease solution” Does the author mean supernatant obtained from the bacterial culture?

Comment 2: How the authors differentiate between the specific activity 2.69 U/mg (Line 219) and 33.3 U/mg (line 231) for the same enzyme?

Comment 3: The quality of Fig 2B &D need improvement as x-axis titles are not clearly readable

Comment 4: If the author mention the Vmax value in U/mg, that will be more appropriate to compare with their specific activity already mentioned in U/mg

Comment 5: The SDS PAGE showing clear degradation of Sp, whole degradation of MP is not conclusive from SDS PAGE, which might be due to overloaded sample resulted in a mesh background. The author can either re-run an SDS PAGE with reducing volume of the sample or perform alternative quantitative analysis will make the study more compact.

Comment 6: The author can provide a future perspective of this enzyme in conclusion.

Comment 7: Overall the manuscript is well written, however, there are a lot of grammatical errors in manuscript. The authors need to pay a careful attention to omit these errors.

Reviewer 2 Report

This is very well designed study on protease from one of the microbial components of Chinese dry sausages - Staphylococcus xylosus A2.The enzyme was isolated, suitably purified and well characterized. In order to check if it could be of use in practice inn meat industry studies on oits action of meat were also done. Thus, paper is a comprehensive description of interesting enzyme, which could be used in meat industry. 

Reviewer 3 Report

However, article is quite interesting, but there are several drawbacks. Authors need to correct those.

  1. Authors used a growth medium but the reference is missing. It is necessary to mention. Also, it is necessary to provide the growth curve of isolated microbe.
  2. Figures are extremely bad quality and sometimes unreadable. It is necessary to improve or increase the size.
  3. In Table 2, several derivative kinetic parameters are represented. It is not required. Ea, Vmax, Km, kcat, Delta H, Delta G, Delta S are enough. 
  4. Protocol mentioned in section 2.16 is not appropriate to describe the concentration of enzyme. I can see that enzyme is not fully purified. It is necessary to use preparative HPLC to prepare fraction with highest enzyme activity and use that result for thermodynamic calculation.
  5. Some results are quite confusing. In table 2, results were evaluated considering casein as a substrate, mentioned in section 2.3. What about table 3 ? How they evaluated ? Why table 2 and 3 are not together ? Detailed methodology for determination of kinetic parameters are required. Confidence level for determination of thermodynamic and kinetic parameters are required. Is 2.3 section describe protease assay ? Need correction.
  6. It is necessary to improve the conclusion section. Summarized information together with future scope, limitation of present investigation need to mention.
  7. Protocl in section 2.11 is not correct. Why trypsin and papin is used together with isolated protease ? Reference is missing. This protocol is not justified.
  8. Why author choose particularly HEK-293 cells ? Did other cell lines considered in investigation ? 

Round 2

Reviewer 1 Report

The author has made all the suggested corrections and answered all the comments. I recommend the manuscript may now be accepted in the current form. 

Reviewer 3 Report

Responses from authors are highly satisfied. I am suggesting to accept the manuscript.